# TPH1 and 5-HT_7_ Receptor Overexpression Leading to Gemcitabine-Resistance Requires Non-Canonical Permissive Action of EZH2 in Pancreatic Ductal Adenocarcinoma

**DOI:** 10.3390/cancers13215305

**Published:** 2021-10-22

**Authors:** Prakash Chaudhary, Diwakar Guragain, Jae-Hoon Chang, Jung-Ae Kim

**Affiliations:** College of Pharmacy, Yeungnam University, Gyeongsan 38541, Korea; prakash@ynu.ac.kr (P.C.); diwakarguragain@ynu.ac.kr (D.G.); jchang@yu.ac.kr (J.-H.C.)

**Keywords:** pancreatic ductal adenocarcinoma, gemcitabine-resistance, cancer stem cells, Enhancer of zeste homolog 2, tryptophan hydroxylase 1, 5-HT_7_

## Abstract

**Simple Summary:**

Most patients with pancreatic cancer initially respond to the first-choice drug gemcitabine, but the cancer cells rapidly acquire drug resistance, resulting in poor survival. In this study, we investigated whether the serotonin (5-hydroxytryptamine, 5-HT) system plays an important role in gemcitabine resistance and the maintenance of pancreatic cancer stem cells (CSCs) in association with an Enhancer of zeste homolog 2 (EZH2), an epigenetic regulator of transcription. Herein, we demonstrate that long-term exposure of PDAC cells to 5-HT leads to enhanced EZH2 expression which, in turn, allows upregulation of TPH1 and 5-HT_7_, resulting in EZH2-TPH1-5-HT_7_ axis operating in a feed-forward manner. The results suggest that the EZH2-TPH1-5-HT_7_ axis may be a highly efficient therapeutic target against drug-resistant pancreatic ductal adenocarcinoma (PDAC).

**Abstract:**

In the present study, we investigated the regulatory mechanisms underlying overexpression of EZH2, tryptophan hydroxylase 1 (TPH1), and 5-HT_7_, in relation to gemcitabine resistance and CSC survival in PDAC cells. In aggressive PANC-1 and MIA PaCa-2 cells, knock-down (KD) of EZH2, TPH1, or HTR7 induced a decrease in CSCs and recovery from gemcitabine resistance, while preconditioning of less aggressive Capan-1 cells with 5-HT induced gemcitabine resistance with increased expression of EZH2, TPH1, and 5-HT_7_. Such effects of the gene KD and 5-HT treatment were mediated through PI3K/Akt and JAK2/STAT3 signaling pathways. EZH2 KD or GSK-126 (an EZH2 inhibitor) inhibited activities of these signaling pathways which altered nuclear level of NF-kB, Sp1, and p-STAT3, accompanied by downregulation of TPH1 and 5-HT_7_. Co-immunoprecipation with EZH2 and pan-methyl lysine antibodies revealed that auto-methylated EZH2 served as a scaffold for binding with methylated NF-kB and Sp1 as well as unmethylated p-STAT3. Furthermore, the inhibitor of EZH2, TPH1, or 5-HT_7_ effectively regressed pancreatic tumor growth in a xenografted mouse tumor model. Overall, the results revealed that long-term exposure to 5-HT upregulated EZH2, and the noncanonical action of EZH2 allowed the expression of TPH1-5-HT_7_ axis leading to gemcitabine resistance and CSC population in PDAC.

## 1. Introduction

Pancreatic ductal adenocarcinoma (PDAC) accounts for over 90% of pancreatic cancer cases worldwide and is one of the most aggressive human malignancies. It is the only cancer type in which the five-year survival rate (8% in metastatic PDAC) has not improved over the last several decades despite advances in surgical and other cancer treatments [1]. Gemcitabine is the first line chemotherapeutic drug for PDAC. However, the rate of response to gemcitabine is poor, and PDAC rapidly develop resistance [2,3].

Genomic studies of PDAC have revealed four frequently mutated genes, *KRAS*, *TP53*, *CDKN2A*, and *SMAD4*. The activating mutation in *KRAS* is present in 80–90% of pancreatic cancers [4], and this mutation results in overactivation of various signaling molecules, such as Raf/MEK/extracellular signal-regulated kinase (ERK) and phosphoinositide 3-kinases (PI3Ks)/Akt [5]. These signaling molecules mediate early distant metastasis, resistance to conventional chemotherapy [6,7], and development of pancreatic cancer stem cell (CSC) characteristics [8]. In addition to the genetic alterations, it is found that aberrant activation of epigenetic regulatory mechanisms, DNA methylation and post-translational histone modifications also play an important role in PDAC progression [9]. The epigenetic alterations are also responsible for over-activation of the growth signaling pathway and the silencing of tumor suppressor and cell cycle checkpoint genes in PDAC [10,11].

Enhancer of zeste homolog 2 (EZH2) is an enzymatic catalytic subunit of polycomb repressive complex 2 (PRC2) that can alter gene silencing by trimethylation of lysine in histone 3 (H3K27me3), a form of chromatin structure modulation [12]. In addition, EZH2 also exerts PRC2-dependent methylation of non-histone proteins and PRC2-independent gene transactivation [13]. Moreover, EZH2 interacts with intracellular signaling molecules. For example, phosphorylation of EZH2 at Ser21 by Akt methylates STAT3, leading to constitutive STAT3 activation [14]. This phenomenon could possibly explain constitutive activation of STAT3 in PDAC cells which overexpress EZH2 [15] and lack JAK2 activation mutation [16]; however, it has not been proven in PDAC yet. In various cancer cells, EZH2 is known to play a critical role in the maintenance [17,18] and expansion [19] of stem cell-like characteristics through activation of stemness-associated signaling pathways. EZH2 not only activates NF-kB through epigenetic silencing of disabled homolog 2-interacting protein (DAB2IP), a GTPase activating protein [20], but also directly binds to NF-kB, leading to activation of target gene expressions [21]. On the other hand, *EZH2* gene expression itself is regulated by NF-kB and other transcription factors [22].

Serotonin (5-hydroxytryptamine, 5-HT), synthesized by tryptophan hydroxylase (TPH) 1 in the periphery and by TPH2 in the central nervous system [23], exerts various physiological and pathophysiological actions including mitogenic action for a wide range of normal and tumor cells [24,25]. Normally, 95% of peripheral 5-HT is synthesized in the enterochromaffin cells, and it is transported and stored in the platelets [23]. The way for most peripheral cells to receive such 5-HT action is either a paracrine method, contacting 5-HT secreted from activated platelets [26,27], or an autocrine method, in which 5-HT is secreted from the cell itself with increased TPH1 expression [24,28]. During gestational pancreatic β-cell proliferation, TPH1 induction and subsequent 5-HT activity through the autocrine-paracrine loop was accompanied by an increase in EZH2 expression [29]. However, the mechanisms underlying their concurrent upregulation have not been elucidated.

Similar to the non-neuronal action of 5-HT, including dedifferentiation of acinar cells and promotion of regeneration after pancreatitis [30], overexpression of 5-HT_1B_ and 5-HT_1D_ receptors stimulates pancreatic cancer progression by promoting proliferation and invasion of PDAC [31]. On the other hand, 5-HT_7_ receptors are known to stimulate proliferation and invasion of breast cancer cells via the PI3K/Akt pathway [28]. However, the linkage of 5-HT receptors to JAK/STAT3 signaling pathway in PDAC remains unclear.

In the present study, we examined the regulatory mechanisms underlying the overexpression of EZH2, TPH1, and 5-HT_7_ in PDAC, in relation to CSCs and gemcitabine resistance. Also, we investigated whether long-term exposure to TPH1-derived or exogenous 5-HT induces pancreatic cancer cells to a gemcitabine-resistant phenotype, and the mode of operation of the TPH1-5-HT-5-HT_7_ axis. Finally, we confirmed a critical role of the EZH2-TPH1-5-HT-5-HT_7_ axis in gemcitabine-resistant pancreatic cancer growth in an in vivo xenograft tumor model.

## 2. Materials and Methods

### 2.1. Materials

Dulbecco’s Modified Eagle’s Medium (DMEM) and RPMI-1640 were obtained from Hyclone (Logan, UT, USA). Keratinocyte serum-free medium, recombinant endothelial growth factor (rEGF), bovine pituitary extract (BPE), fetal bovine serum (FBS), penicillin, and streptomycin were obtained from Gibco (Grand Island, NY, USA). Stattic and gemcitabine were purchased from Tocris (Bristol, UK), wortmannin from Biomol International (Plymouth Meeting, PA, USA), Trizol from Life Technologies Inc. (Carlsbad, CA, USA). Fedratinib, SB-269970, telotristat was purchased from Selleckchem (Houston, TX, USA), and GSK-126 from MedChemExpress (Princeton, NJ, USA). Antibodies against p-PI3K, PI3K, p-Akt, Akt, p-JAK2, JAK2, p-STAT3, STAT3, EZH2, and NF-κB P65 were purchased from Cell Signaling Technology Inc. (Beverly, MA, USA); 5-HT_1A_, 5-HT_1B_, Nanog, CD44, Sp1, and EZH1 were from Abcam (Cambridge, MA, USA); TPH1 from Invitrogen (Carlsbad, CA, USA); β-Actin and Lamin B from Santa Cruz Biotechnology (Dallas, Texas, USA). FITC-anti-human CD44, APC-anti-human CD24, PE-anti-mouse IgG2B, FITC-anti-mouse IgG1 kappa isotype control, APC-anti-mouse IgG2a kappa isotype control antibodies were purchased from Biolegend (San Diego, CA, USA). Two types of anti-5-HT_7_ antibodies were purchased from Novus Biologicals (Littleton, CO, USA) and R&D system (Minneapolis, MN, USA) for immunoblotting and flow cytometry, respectively.

### 2.2. Cell Culture and Viability Assay

Human PDAC cell lines (PANC-1, MIA PaCa-2, Capan-1, and Capan-2) and normal pancreatic ductal epithelial cell lines (H6c7) were purchased from the Korean Cell Line Bank (Seoul, South Korea) and Kerafast (Boston, MA, USA), respectively. PDAC cell lines were cultured in DMEM (PANC-1 and MIA PaCa-2) or RPMI-1640 medium (Capan-1 and Capan-2) supplemented with 10% FBS and 1% penicillin/streptomycin. H6c7 cells were maintained in keratinocyte serum-free medium supplemented with rEGF and BPE. All the cells were incubated at 37 °C under a 5% CO2. For measurement of cell viability, cells were seeded in 96-well plate in 1% FBS containing media. After cells were treated with vehicle or drugs for 48 h, 3-(4,5-dimethylthiazol-2-yl)-2,5-diphenyltetrazolium bromide (MTT) (Merck, Burlington, MA, USA) dye solution was added. After 4 h, the solution was removed, and dimethyl sulfoxide was added to dissolve the formazan crystal. The absorbance was measured at 540 nm using a microplate reader (Versamax, Molecular Devices, Inc., San Jose, CA, USA).

### 2.3. Protein Extraction and Western Blotting

Cells were lysed using the radioimmunoprecipitation assay (RIPA) buffer (Thermo Scientific, Waltham, MA, USA) containing 1× protease and phosphatase inhibitor cocktail (Thermo Scientific) for total protein extraction. Nuclear and cytoplasmic proteins were extracted using NE-PER Nuclear and cytoplasmic extraction reagent, respectively (Thermo Scientific). Proteins separated by SDS-PAGE were transferred onto nitrocellulose membrane (Whatman GmbH, Dassel, Germany), and immunoblotted with specific primary and secondary antibodies. The immunoblots were visualized using an ECL kit (Thermo Scientific) and imaged using LAS-4000 mini system (Fuji, Tokyo, Japan).

### 2.4. siRNA Transfection

Cells were seeded in antibiotic-free DMEM high glucose media and transfected using MISSION esiRNAs (100 nM) targeting *HTR7*, *TPH1*, *EZH1* or *EZH2* (Sigma-Aldrich, St Louis, MO, USA) using DharmFECT reagent 4 (Thermo Scientific) for 72 h. The esiRNAs are comprised of a heterogeneous pool of siRNA (natural RNA, no modifications) that all target the same mRNA sequence, so that these multiple silencing triggers lead to highly specific and effective gene knockdowns with lower off-target effects than single, chemically-synthesized siRNA.

### 2.5. Flow Cytometry Analysis

Single cell suspension (1 × 10^7^ cells/mL) in cold PBS containing 3% FBS were stained with anti-human 5-HT_7_ or isotype control antibodies for 30 min, followed by PE-anti-mouse IgG2B-secondary antibody, APC-anti-human CD24, and FITC-anti-human CD44 for 30 min in the dark at 4 °C. Stained cells were washed twice and analyzed by flow cytometry (FACSVerse Cytometer, BD Biosciences, San Jose, CA, USA).

### 2.6. Quantitative Real-Time Polymerase Chain Reaction (qRT-PCR)

Trizol-extracted total RNA was converted to cDNA using the GoScript reverse transcription system (Promega Corporation, WI, USA). mRNA levels were quantified using a QuantiTect SYBR Green PCR kit (Qiagen, Valencia, CA, USA). Primers were obtained from Bioneer Corporation (Daejeon, South Korea). The primer sequences used were *TPH1* (sense 5′-GCCAGTCATCCAGGAACATT-3′ and anti-sense 5′-ATTGTTTGGCCAGAAGATGC-3′), *HTR7* (sense 5′-TGAGTCTAGGCGTTGTGGTG-3′ and anti-sense 5′-TGCTTGGAAAAGCCTTCTGT-3′), *EZH2* (sense 5′-TTGTTGGCGGAAGCGTGTAAAATC-3′ and anti-sense 5′-TCCCTAGTCCCGCGCAATGAGC-3′), and internal control *GAPDH* (sense 5′-ACCACAGTCCATGCCATCAC-3′ and anti-sense 5′-TCCACCACCCTGTTGCTGTA-3′).

### 2.7. Sphere Formation Assay

One thousand cells were seeded on an ultra-low adhering 24-well plate (Corning Incorporated Costar, Corning, NY, USA) in prEGM media (Lonza, Basel, Switzerland) and allowed to form spheres. After four days, the spheres were treated with vehicle or chemical inhibitors (1 and 3 μM). After 11 days of drug treatment, images of spheres were captured using an inverted microscope (IX73, Olympus, Tokyo, Japan). The number of spheroids over 50 μm in diameter was counted by using Image J 1.48v software (National Institute of Health, Bethesda, MD, USA).

### 2.8. Immunoprecipitation

Total and nuclear proteins (100 μg) were immunoprecipitated with IP-grade EZH2, pan methyl lysine antibody (1 mg/mL), or IgG (1 mg/mL) (Sigma-Aldrich) for 16 h at 4 °C. Protein A agarose beads (50 μL) (Thermo Scientific) was added in the immunoprecipitated solution for 1 h at 4 °C. Then, the immune complexes were collected after centrifugation at 3000× *g*, 2 min at 4 °C. The pellet was washed with PBS (twice), re-suspended in 25 μL of 1× sample buffer (62.5 mM Tris-HCl pH 6.8, 2.5% SDS, 0.002% Bromophenol Blue, 0.7135 M (5%) β-mercaptoethanol, 10% glycerol), and heated at 95 °C for 5 min. After centrifugation at 12,000× *g*, 30 s at 4 °C, supernatant (IP samples) was collected.

### 2.9. Anti-Tumor Activity Measurement Using a Xenograft Tumor Model

Female BALB/c nude mice (OrientBio, Gyeonggi, South Korea) were subcutaneously inoculated with 1 × 10^7^ PANC-1 cells/Matrigel (1:1) at the right flank. After tumor volume reached approximately 300 mm^3^, mice in the first set of experiments were administered intraperitoneally (i.p.) with drugs (gemcitabine (50 mg/kg), telotristat (1 or 10 mg/kg), SB-269970 (1 or 10 mg/kg), or GSK-126 (10 mg/kg)) once a day for six days a week (*n* = 5). In a different set of experiments, a mouse tumor model was made with the same method as before, except the number of mice in each group (*n* = 6). In this second set of experiments, mice were administered with gemcitabine (50 mg/kg), GSK-126 (10 mg/kg), or gemcitabine plus drugs (telotristat (10 mg/kg), SB-269970 (10 mg/kg), or GSK-126 (10 mg/kg)). Tumor volume was calculated using the equation (l × b^2^)/2, where l and b were the larger and smaller dimensions of each tumor. On the final day of treatment, tumors were excised from sacrificed mice by CO_2_ gas inhalation, and tumor weight was measured.

The mouse experiments were performed following the institutional guidelines of the Institute of Laboratory Animal Resources and approved by the Institutional Animal Care and Use Committee of Yeungnam University.

### 2.10. Statistical Analyses

Data from more than three independent experiments were averaged and expressed as mean ± SEM. Statistical significance was determined with the one-way analysis of variance (ANOVA), followed by the Newman-Keul’s comparison method, using Graph Pad Prism 5.0 (San Diego, CA, USA). *p*-values less than 0.05 were considered statistically significant.

## 3. Results

### 3.1. EZH2 Supports TPH1-5-HT_7_ Axis to Regulate Gemcitabine Resistance and Cancer Stem Cell Population in Pancreatic Cancer Cells

To investigate the relationship between intrinsic gemcitabine resistance and the expression of EZH2 and 5-HT system genes in pancreatic cancer cells, we first compared the levels of resistance and those gene expressions between PDAC cell lines. H6c7 was used as a negative control cell line for gemcitabine resistance (GemR). PANC-1 cells exhibited the strongest gemcitabine resistance, followed by MIA PaCa-2, Capan-1, and Capan-2 cells (Figure 1A). Corresponding to GemR levels, the level of EZH2, TPH1 and 5-HT_7_ receptor expressions was the highest in PANC-1, followed by MIA PaCa-2, Capan-1, and Capan-2 (Figure 1B and Appendix A), whereas EZH1, 5-HT_1A_, and 5-HT_1B_ levels did not correlate with GemR levels, and they were higher in Capan-1 and Capan-2 cells than PANC-1 and MIA PaCa-2 cells (Figure 1B). Treatment of PANC-1 and MIA PaCa-2 cells with GSK-126, an EZH2 inhibitor (Figure 1C and Appendix A), or *EZH2* knock-down (KD) with siRNA transfection (Figure 1D and Appendix A) down-regulated the expression of TPH1 and 5-HT_7_. In addition, KD of TPH1 or 5-HT_7_ induced suppression of each other’s expression without changes in EZH2 mRNA (Appendix A) and protein (Figure 1E) levels. Down-regulation of these genes with siRNA induced recovery from gemcitabine resistance in PANC-1 and MIA PaCa-2 cells (Figure 1F). On the other hand, prolonged exposure of H6c7 and Capan-1 cells to 5-HT (10 μM, 96 h) induced gemcitabine resistance (Figure 1G), accompanied by up-regulation of EZH2, TPH1, and 5-HT_7_ expressions (Figure 1H). The results indicate that EZH2 acted as an upstream regulator of the TPH1-5-HT_7_ axis to maintain gemcitabine resistance of PDAC cells. In addition, prolonged exposure of Capan-1 cells to 5-HT significantly enhanced sphere formation in Capan-1 cells (Figure 1I).

We then investigated whether EZH2-TPH1-5-HT_7_ axis regulates the pancreatic CSC population, which is responsible for chemo-resistance and cancer relapse [32], using FACS analysis and sphere forming ability. The CSC population in each cell line detected with antibodies against pancreatic CSC markers, CD24 and CD44 [18,33], was 5.42, 5.44, 0.83, and 0.05% in PANC-1, MIA PaCa-2, Capan-1, and Capan-2, respectively (Figure 2A). The CSC population in PANC-1 and MIA PaCa-2 cells were significantly reduced by silencing TPH1, 5-HT_7_, or EZH2 (Figure 2B). In PANC-1 cells, the CD24/CD44-positive CSC population was mostly positive for 5-HT_7_ (Figure 2C), whereas the 5-HT_7_-positive CSC population (CD24/CD44/5-HT_7_) in MIA PaCa-2 cells was much smaller than CD24/CD44-double positive population (Figure 2D). However, both cell lines responded similarly to the KD of the EZH2-TPH1-5-HT_7_ axis gene, with a decrease in the triple-positive (CD24/CD44/5-HT_7_) population (Figure 2C,D). Interestingly, the inhibitory effect of EZH2 KD on 5-HT_7_ level in CSCs was much greater than that of either 5-HT_7_ or TPH1 KD, indicating that regulatory action of EZH2 in 5-HT_7_ expression involved mechanisms other than the signaling associated with TPH1 and 5-HT_7_.

We then examined whether the EZH2-TPH1-5-HT_7_ axis was linked to PI3K/Akt and JAK2/STAT3 signaling pathways, which are associated with drug resistance [6,7] and pancreatic CSC characteristics [8]. In PANC-1 and MIA PaCa-2 cells, the phosphorylation of PI3K/Akt and JAK2/STAT3 were suppressed by KD of EZH2-TPH1-5-HT_7_ axis gene (Figure 3A) and GSK-126 treatment (Figure 3B).

We further investigated the relative contribution of PI3K/Akt and JAK2/STAT3 pathways in the maintenance of the CSC population by measuring sphere-forming ability of the cells in the presence of the signaling molecule inhibitors. At fixed concentrations (1 and 3 μM) which were selected based on cell viability response to the inhibitors (Appendix A), gemcitabine did not inhibit sphere formation of PANC-1 and MIA PaCa-2 cells, whereas inhibitors of PI3K/Akt, JAK2/STAT3, and EZH2/TPH1/5-HT_7_ axis significantly suppressed the sphere formation of PANC-1 and MIA PaCa-2 in a concentration-dependent manner (Figure 4A,B). In both PANC-1 and MIA PaCa-2 spheres, the expressions of CD44, a CSC surface marker, and Nanog, a stemness-associated transcription factor (TF), were most significantly suppressed by GSK-126, followed by telotristat (TPH1 inhibitor), fedratinib (JAK2 inhibitor), stattic (STAT3 inhibitor), wortmannin (PI3K inhibitor), and SC-66 (Akt inhibitor) (Figure 4C and Appendix A). Interestingly, EZH2 expression in the spheres was down-regulated by the inhibitors except telotristat and SB-269970, whereas TPH1 and 5-HT_7_ expressions in the spheres were suppressed by all the inhibitors in both PANC-1 and MIA PaCa-2 (Figure 4C and Appendix A). The results indicate that although PI3K/Akt and JAK2/STAT3 signaling pathways were commonly involved in the expression of EZH2, TPH1, and 5-HT_7_, the regulatory mechanism for the expression and action of EZH2 was different from that for the TPH1-5-HT_7_ axis in PDAC cells. To reveal this difference, we examined whether the gene transcriptional repression action of EZH2 is linked to these signaling pathways. We found that EZH2 KD increased expression of DAB2IP and PTEN (Figure 4D), which are known to inhibit Ras and PI3K, respectively [34,35,36,37].

### 3.2. EZH2-Regulated Signaling Pathways Potentiate Nuclear Translocation of TFs Linked to TPH1-5-HT_7_ Axis in Pancreatic Cancer Cells

We also examined which TFs were responsible for the upregulation of EZH2, TPH1, and 5-HT_7_ expressions. In PANC-1 and MIA PaCa-2 cells, EZH2, TPH1, and 5-HT_7_ expressions were down-regulated by treatment with mithramycin A (Sp1 inhibitor), PDTC (NF-κB inhibitor), and stattic (STAT3 inhibitor), but not by SR11302 (AP-1 inhibitor) and KG-501 (CREB inhibitor) (Figure 5A), suggesting Sp1, NF-κB, and STAT3 were involved in these gene expressions. In an inverse proportion to the extent of EZH2 decrease by the TF inhibitors, DAB2IP and PTEN protein levels were increased by treatment with the TF inhibitors (Figure 5A). The nuclear levels of Sp1, NF-κB, and p-STAT3 were inhibited by gallein (Gβγ inhibitor) (Figure 5B). However, nuclear Sp1 and NF-κB levels were more suppressed by PI3K/Akt inhibitors than by JAK2/STAT3 inhibitors, while nuclear p-STAT3 level was reduced in the opposite direction by those inhibitors (Figure 5B). Moreover, telotristat and SB-269970 reduced the nuclear p-STAT3 level, but not the NF-κB or Sp1 level (Figure 5B). In 5-HT-pretreated Capan-1 cells, similar regulatory actions of the inhibitors of signaling molecule (Figure 5C) and TFs (Figure 5D) in the expression of EZH2, TPH1 and 5-HT_7_ were observed.

### 3.3. Automethylated EZH2 Serves as a Binding Scaffold for Methylated NF-κB and Sp1, and Unmethylated p-STAT3, in a PRC2-Independent Manner

To further identify whether EZH2 acted as a scaffold for NF-κB, STAT3, and Sp1 to up-regulate EZH2-TPH1-5-HT_7_ gene expressions, we performed co-IP experiment with anti-EZH2 antibody. Co-precipitation of EZH2 with nuclear NF-κB was found in PANC-1 (Figure 6A), MIA PaCa-2, and Capan-1 cells (Figure 6B), and the binding was further enhanced by 5-HT treatment (Figure 6B). Similarly, binding of EZH2 with Sp1 and STAT3 was observed, and such binding was further increased by 5-HT treatment (Figure 6B). However, SUZ12, a component of PRC2, was not precipitated with EZH2 (Figure 6B), indicating that such scaffold action of EZH2 was PRC2-independent. We also examined whether transactivation ability of EZH2 was associated with auto-methylation activity of EZH2, which induces self-activation and methylation of other proteins [38]. In the total protein co-immunoprecipitates with anti-pan methyl lysine antibody, EZH2 and NF-κB were highly methylated, and Sp1 methylation was relatively low level, whereas signaling molecules, PI3K, Akt, JAK2, and STAT3 were not methylated in both PANC-1 and MIA PaCa-2 cells (Figure 6C). In the nuclear protein precipitates with anti-pan methyl lysine antibody, it was confirmed that nuclear STAT3 and p-STAT3 were not methylated (Figure 6D). Moreover, EZH2 KD or GSK-126 treatment significantly suppressed the methylation of nuclear NF-κB and Sp1, in addition to blocking methylation of EZH2 itself (Figure 6D).

### 3.4. Antitumor Effects of EZH2-TPH1-5-HT_7_ Axis Inhibition in PANC-1 Xenograft Tumor Model in Mice

Next, we confirmed that the EZH2-TPH1-5-HT_7_ axis is a useful therapeutic target against drug-resistant pancreatic cancer, using an in vivo tumor model in which PANC-1 cells were subcutaneously transplanted. Compared to the vehicle-treated control group, treatment with gemcitabine (50 mg/kg) induced a slight decrease in PANC-1 tumor size, whereas telotristat (1 or 10 mg/kg) and SB-269970 (1 or 10 mg/kg) significantly reduced tumor growth in a dose-dependent manner (Figure 7A,B). In addition, telotristat and SB-269970 at a dose of 10 mg/kg started regressing tumor-growth on day 31 of drug treatment (Figure 7A,C). Throughout the treatment period of 42 days, the body weight of drug (telotristat or SB-269970)-treated mice was not significantly different from that of control mice (Figure 7D). In a separate set of experiments, treatment with GSK-126 (10 mg/kg) showed a similar response, and tumor size started to regress on day 31 of treatment (Figure 7E,F). The tumor regression effect of GSK-126 plus gemcitabine was not significantly different from that of GSK-126 alone (Figure 7E). Similarly, the effect of co-administration of telotristat or SB-269970 with gemcitabine was not different from that of combinated treatment of GSK-126 with gemcitabine (Figure 7E). However, the tumor weight in the group co-administered with GSK-126 and gemcitabine was significantly lower than that of the GSK-126 only group. In addition, the effect of co-administration of other drugs and gemcitabine was not significantly different from that of combinated treatment of GSK-126 and gemcitabine (Figure 7G). Body weight of mice co-treated with gemcitabine and the other drugs was not significantly different from that of control mice until treatment day 31. Thereafter, the body weight of GSK-126 alone mice or the combinated treatment group was significantly decreased compared to that of control group (Figure 7H).

## 4. Discussion

It has been reported EZH2 and 5-HT derived from peripheral TPH1 in cancer tissues independently contributes to cancer malignancy by stimulating cancer cell proliferation and invasion [28,39,40,41]. The present study demonstrated for the first time that EZH2 permits up-regulation of 5-HT system genes, TPH1 and 5-HT_7_, leading to drug resistance and CSC maintenance in PDAC. In addition, we also revealed that such noncanonical EZH2 action is mediated through the PI3K/Akt and JAK2/STAT3 pathways in a feed-forward manner in association with 5-HT_7_.

Previous studies on pancreatic cancer cell lines and human tumor tissue microarray have shown the role of 5-HT_1B_, 5-HT_1D_, or 5-HT_2B_ in 5-HT-induced cancer cell proliferation and metabolism [31,39,42]. However, the most recently discovered 5-HT receptor, 5-HT_7_, is increasingly reported as a new therapeutic target to inhibit cancer proliferation, migration, and invasion [28,42,43,44]. The current study demonstrated an additional important role of 5-HT_7_ in inducing drug resistance and CSCs in pancreatic cancer by elucidating the mechanism regulating 5-HT_7_ expression: that is, inhibition of 5-HT_7_ by its antagonist or removal of 5-HT via TPH1 inhibition suppressed 5-HT_7_ expression in PANC-1 and MIA PaCa-2 cells. Similarly, TPH1 expression was also inhibited by a TPH1 inhibitor and 5-HT_7_ antagonist. Such mutual regulation of TPH1 and 5-HT_7_ expression via 5-HT in PDACs suggests that TPH1-5-HT-5-HT_7_ axis operates in a feed-forward manner in PDAC cell lines. In addition, in Capan-1 cells expressing very low level of TPH1 and 5-HT_7_, 5-HT treatment up-regulated TPH1 and 5-HT_7_ expressions accompanying gemcitabine resistance. Notably, EZH2 inhibition down-regulated EZH2 itself, TPH1 and 5-HT_7_ expression, whereas inhibition of TPH1-5-HT-5-HT_7_ axis reduced expression of TPH1 and 5-HT_7_, but not of EZH2. The results indicate that EZH2 acts as a master regulator permitting the expression of TPH1 and 5-HT_7_, forming EZH2-TPH1-5-HT-5-HT_7_ axis.

EZH1, a paralog of EZH2, is widely expressed in non-proliferating adult cells, whereas EZH2 is preferentially expressed in proliferating cells [45]. EZH1 complements EZH2 in maintaining stem cell identity and executing pluripotency [46]. In the current study, we found EZH2 KD with esiRNA slightly decreased EZH1 expression levels, although we used EZH2 esiRNA with guaranteed specificity and effectiveness. The result may be due to their 63% sequence homology and the 94% identity of their SET domain. This result suggests that it will be important to use a highly specific silencing strategy and to perform rescue experiments in a future study. In addition, a recent study reported that EZH1 is globally distributed in the chromatin of aggressive lymphomas, and both EZH1 and EZH2 play critical roles in the chromatin regulation [47]. Given the report, a future study is also required to determine whether EZH1 also contributed to the modulation of TPH1 and 5-HT_7_ pathways in pancreatic cancer cells.

The operation of TPH1-5-HT-5-HT_7_ axis utilized PI3K/Akt and JAK2/STAT3 signaling pathways that were activated through Gβγ components of the receptor, consistent with our previous findings in breast cancer cells [28]. There is a report that indicates the JAK2 signaling pathway is linked with TPH1 induction by activation of prolactin receptor, a cytokine receptor, in pancreatic β cells [29,48]. Here, the current study first reported that JAK2/STAT3 is linked to 5-HT_7_ receptor in PDAC cells, in addition to the PI3K/Akt signaling pathway. We also demonstrated that 5-HT-enhanced EZH2 also regulated PI3K/Akt and JAK2/STAT3 signaling pathways. In addition, the fact that PI3K/Akt activity was dependent not only on the Gβγ component but also on Ras, which is overactivated by gain-of-function mutation in PDAC, was confirmed by the results that 5-HT_7_ KD or treatment with inhibitor (SB-269970) maintained PI3K/Akt-dependent NF-κB nuclear translocation and EZH2 expression level. Moreover, our study confirmed that EZH2 KD, but not TPH1-5-HT_7_ KD, induced an increase in the expression of DAB2IP and PTEN, which inhibit Ras and PI3K, respectively [34,35,36,37]. These results demonstrate that once EZH2 is expressed, it performs a master regulatory action on intracellular signaling pathways through two directions, (1) activation of PI3K/Akt and JAK2/STAT3 through transactivation of TPH1-5-HT_7_ expressions, and (2) supporting PI3K/Akt activity by inhibition of the signals that inhibit PI3K/Akt through downregulation of DAB2IP and PTEN (Figure 8). Despite the present results with PDAC cell lines, future studies will be needed to further evaluate the molecular pathways of the EZH2-TPH1-5-HT_7_ axis in vivo.

As confirmed by co-immunoprecipitation with EZH2 and pan methyl lysine antibodies, the trans-activating action of EZH2 was performed in two sequential processes. That is, (1) methylation of EZH2 itself and TFs (NF-kB and Sp1), and (2) methylated EZH2 binds to TFs (NF-kB, Sp1, and p-STAT3), regulating gene expressions. The automethylation of EZH2 is reported as a self-activating mechanism for PRC2 [38]. However, in the current study we found that SUZ12 was not binding with EZH2, indicating the PRC2-independent action of EZH2 in PDAC as reported in breast cancer [49]. Interestingly, although nuclear p-STAT3 was involved in EZH2 transactivating action, p-STAT3 as well as STAT3 were not methylated in PDAC, which is different from other cancer cell types that methylated STAT3 by EZH2 is involved in tumorigenesis of glioblastoma stem-like cells [14].

Overall, EZH2 permits transactivation of TPH1 and 5-HT_7_, allowing the TPH1-5-HT-5-HT_7_ axis to stimulate PI3K/Akt and JAK2/STAT3 pathways in a feed-forward manner in PDAC, leading to maintenance of pancreatic CSC populations and drug resistance. The transactivation of TPH1 and 5-HT_7_ was further reinforced by removal of the PI3K/Akt inhibitory signals, such as DAB2IP and PTEN. The critical role of the EZH2-TPH1-5-HT-5-HT_7_ axis was confirmed in an in vivo mouse tumor model showing similar tumor regressing effects of GSK-126, telotristat, and SB-269970.

## 5. Conclusions

In pancreatic cancer, long-term exposure to 5-HT in an autocrine or paracrine manner induced PRC2-independent EZH2 action that supported the TPH1-5-HT_7_ axis, leading to gemcitabine resistance and a CSC population increase. The inhibition of the EZH2-TPH1-5-HT-5-HT_7_ axis was effective in regressing gemcitabine-resistant pancreatic cancer growth in vivo, suggesting that the axis may be a potential therapeutic target for the treatment of drug-resistant PDAC.

## Figures and Tables

**Figure 1 cancers-13-05305-f001:**
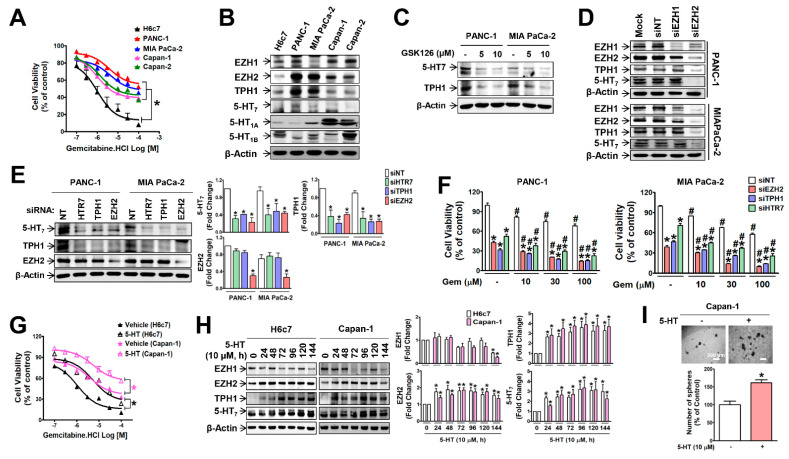
Expression levels of EZH2, TPH1, and 5-HT_7_ correlate with gemcitabine resistance in pancreatic cancer cells. (**A**) Sensitivity of PANC-1, MIA PaCa-2, Capan-1, Capan-2, and H6c7 cells to gemcitabine was determined by measuring cell viability after gemcitabine treatment for 48 h. * *p* < 0.05 compared to H6c7 cells. (**B**) Expression levels of EZH1, EZH2, TPH1, and 5-HT receptors (5-HT_7,_ 5-HT_1A,_ 5-HT_1B_) in PANC-1, MIA PaCa-2, Capan-1, Capan-2, and H6c7. (**C**) Treatment of PANC-1 and MIA PaCa-2 cells with GSK-126 for 48 h inhibited 5-HT_7_ and TPH1 expressions. (**D**) Down-regulation of 5-HT_7_, TPH1, EZH1 and EZH2 expression by EZH2 siRNA (100 nM) in PANC-1 and MIA PaCa-2 cells. NT represents non-target. (**E**,**F**) After transfection of PANC-1 and MIA PaCa-2 cells with HTR7, TPH1, or EZH2 siRNA, expression levels of 5-HT_7_, TPH1, and EZH2 (**E**) and gemcitabine sensitivity (**F**) were measured. * *p* < 0.05 compared to non-target siRNA (siNT)-transfected group. ^#^
*p* < 0.05 compared to the vehicle-treated control. (**G**,**H**) H6c7 and Capan-1 cells turned out to be gemcitabine-resistant phenotype by treatment with 5-HT (10 μM) for 96 h (**G**). Time-dependent treatment with 5-HT up-regulated 5-HT_7_, TPH1, and EZH2 expression (**H**). (**I**) Sphere forming ability of Capan-1 cells pretreated with 5-HT (10 μM) for 96 h was measured. Images of spheres were captured after 14 days of culture in the serum-free medium. Scale bar (white colored) represents 200 μm at original magnification of 4×. The number of spheroids over 50 μm in diameter was counted. * *p* < 0.05, compared to untreated control group. The uncropped Western Blot images can be found in Appendix A.

**Figure 2 cancers-13-05305-f002:**
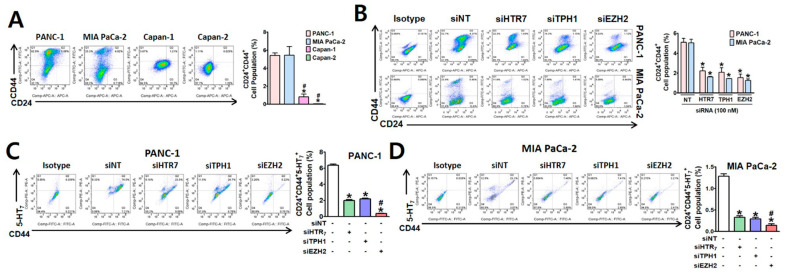
Effects of silencing 5-HT_7_, TPH1 and EZH2 on pancreatic CSC population. (**A**) PANC-1, MIA PaCa-2, Capan-1, and Capan-2 cells stained with antibodies specific to pancreatic CSC surface markers, CD44-FITC and CD24-APC, were analyzed by FACS. Bar graph indicates relative number of CSCs population (CD24^+^CD44^+^) determined from three independent experiments. (**B**) Relative number of CD24^+^CD44^+^ population after KD of HTR7, TPH1, or EZH2 in PANC-1 and MIA PaCa-2 cells. (**C**,**D**) The relative CSC population expressing CD24, CD44, and 5-HT_7_ in PANC-1 (**C**) and MIA PaCa-2 (**D**) cells after KD of HTR7, TPH1, and EZH2. * *p* < 0.05, compared to siNT-transfected group. ^#^
*p* < 0.05, compared to siHTR7- or siTPH1-transfected group.

**Figure 3 cancers-13-05305-f003:**
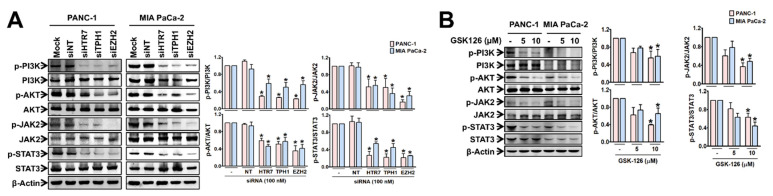
Effects of EZH2-TPH1-5-HT_7_ axis inhibition on the activation of PI3K, Akt, JAK2, and STAT3 in PANC-1 and MIA PaCa-2 cells. (**A**) Cells were treated with siRNA of HTR7, TPH1, and EZH2. * *p* < 0.05, compared to siNT-transfected group. (**B**) Cells were treated with GSK-126 for 48 h. * *p* < 0.05, compared to vehicle-treated control group. The uncropped Western Blot images can be found in Appendix A.

**Figure 4 cancers-13-05305-f004:**
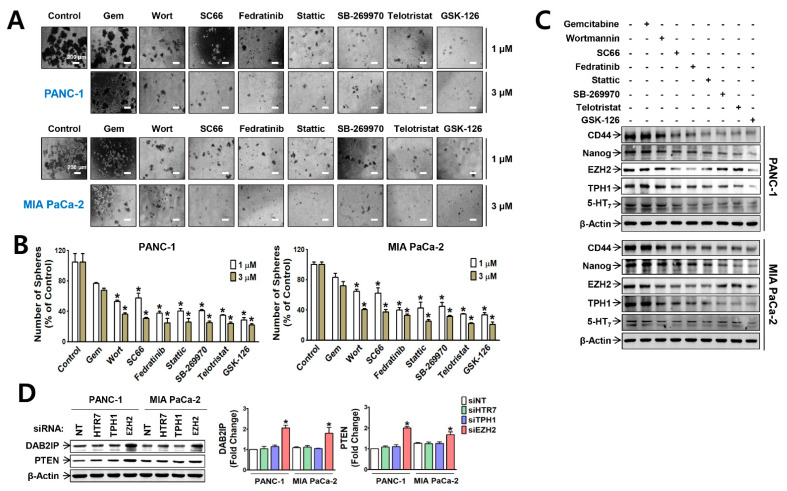
Effects of signaling inhibitors and EZH2-TPH1-5-HT_7_ KD on sphere formation and expression of DAB2IP and PTEN in PANC-1 and MIA PaCa-2 cells. (**A**–**C**) PANC-1 and MIA PaCa-2 cells were treated with 1 or 3 μM of gemcitabine, wortmannin, SC66, fedratinib, stattic, SB-269970, telotristat, and GSK-126 for 11 days, and measured sphere formation. Images of spheres captured after 11 days of the inhibitor treatment (**A**). Scale bar (white colored) represents 200 μm at original magnification of 4×. The number of spheroids over 50 μm in diameter were counted (**B**). * *p* < 0.05 compared to the vehicle-treated control group. Immunoblots of CD44, Nanog, EZH2, TPH1, and 5-HT_7_ from the 1 μM of each inhibitor-treated spheres (**C**). * *p* < 0.05 compared to the vehicle. (**D**) DAB2IP and PTEN expression in the cells treated with siRNA of EZH2, TPH1, or 5-HT_7_ gene. * *p* < 0.05, compared to siNT-transfected group. The uncropped Western Blot images can be found in Appendix A.

**Figure 5 cancers-13-05305-f005:**
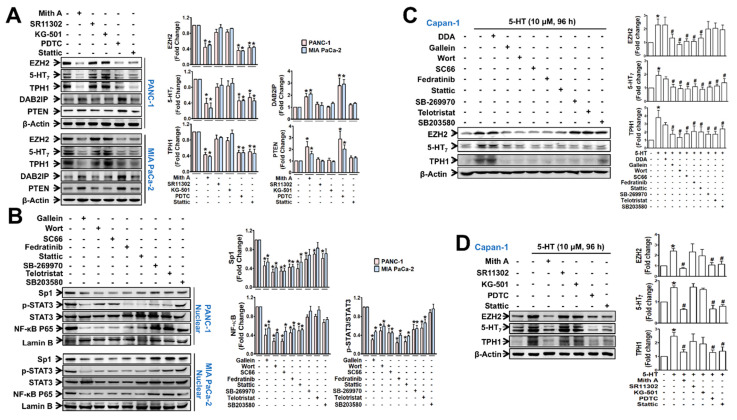
Transcription factors, NF-κB, SP-1, and p-STAT3, are associated with PI3K/Akt and JAK2/STAT3 signaling, leading to 5-HT_7_, TPH1 and EZH2 expression. (**A**) Immunoblots of 5-HT_7_, TPH1 and EZH2 in PANC-1 and MIA PaCa-2 cells treated with mithramycin A (Mith A, 0.1 μM), SR11302 (10 μM), KG-501 (10 μM), PDTC (10 μM), or stattic (10 μM) for 24 h. (**B**) Nuclear levels of SP-1, p-STAT3, and NF-κB (p65) in PANC-1 and MIA PaCa-2 cells treated with gallein (10 μM), wortmannin (3 μM), SC66 (3 μM), fedratinib (3 μM), stattic (3 μM), SB-269970 (3 μM), telotristat (3 μM), or SB203580 (10 μM) for 48 h. Immunoblots are the representative of three independent experiments, and results are the mean ± SEM in the bar diagram. * *p* < 0.05 compared to the vehicle-treated control group. (**C**,**D**) Immunoblots of 5-HT_7_, TPH1 and EZH2 expression in Capan-1 cells. Cells were pretreated with 5-HT (10 μM) for 96 h prior to the treatment with signaling molecule inhibitors for 48 h (**C**) and with transcription factor inhibitors for 24 h (**D**). The bar graphs represent the means ± SEM from three independent experiments. * *p* < 0.05 compared to the vehicle-treated control. ^#^
*p* < 0.05 compared to the 5-HT-treated group. The uncropped Western Blot images can be found in Appendix A.

**Figure 6 cancers-13-05305-f006:**
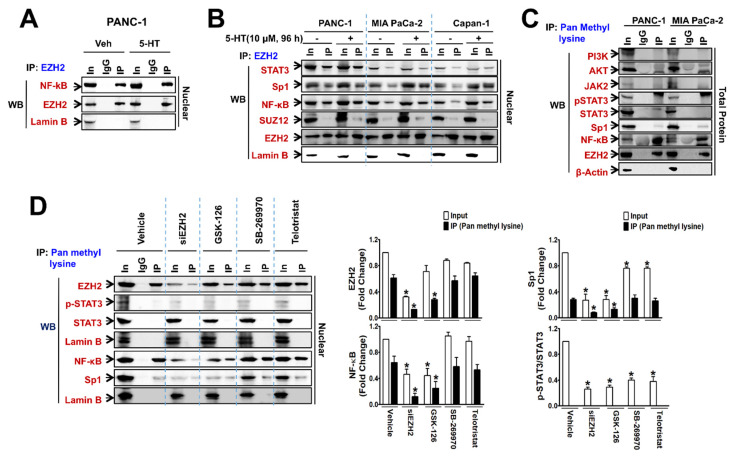
EZH2 binds to TFs, and methylates Sp1 and NF-κB. Nuclear and total proteins extracted from PANC-1, MIA PaCa-2, and Capan-1 cells. For IP-immunoblotting data, antibodies used for co-immunoprecipitation (IP) and western blotting (WB) were labeled as blue and red, respectively. (**A**) Co-IP of NF-κB and EZH2 in PANC-1 cells treated with 5-HT (10 μM) for 96 h. IgG represents a control antibody used for IPs. The In represents input control. Prior to carrying out the IP experiments, cell lysates were subjected to the respective WB as input controls. (**B**) Co-IP of STAT3, Sp1, NF-κB, SUZ12, and EZH2 in PANC-1, MIA PaCa-2, and Capan-1 cells treated with or without 5-HT (10 μM) for 96 h. (**C**) Co-IP and immunoblot analysis of PI3K, AKT, JAK2, p-STAT3, STAT3, Sp1, NF-κB, and EZH2 in PANC-1 and MIA PaCa-2 cells at basal level. (**D**) PANC-1 cells were treated with EZH2 siRNA (100 nM), GSK-126 (10 μM), SB-269970 (3 μM), or telotristat (3 μM) for 48 h. Cell lysates were immunoprecipitated with pan methyl lysine antibody and immunoblotted with pSTAT3, STAT3, EZH2, Sp1, or NF-κB antibody. The bar graphs represent the mean ± SEM of relative density of each protein from three independent experiments. * *p* < 0.05 compared to the vehicle-treated control group. The uncropped Western Blot images can be found in Appendix A.

**Figure 7 cancers-13-05305-f007:**
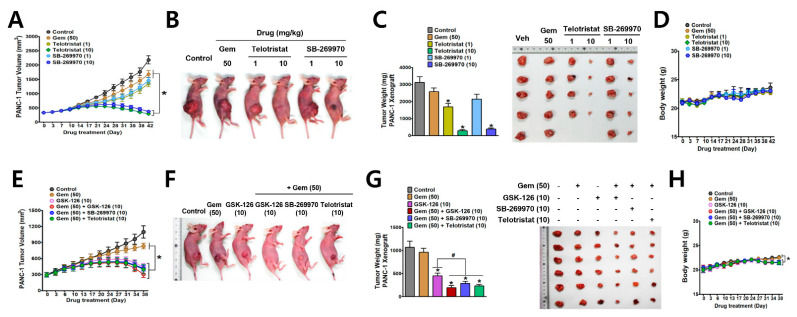
Anti-tumor activities of inhibitors of EZH2-TPH1-5-HT_7_ axis in mouse tumor xenograft models with PANC-1 cells. (**A**–**D**) BALB/c nude mice xenografted with PANC-1 cells were administered i.p. with gemcitabine (50 mg/kg), telotristat (1 and 10 mg/kg), SB-269970 (1 and 10 mg/kg) for 42 days (6 days per week) (*n* = 5). Tumor volume (mm^3^) was measured twice a week (**A**). Representative images showing gross appearance of tumor in mice (**B**) and weight of excised tumor at the time of sacrifice (**C**). Mouse body weight during the drug administration period (**D**). * *p* < 0.05 compared to the vehicle-treated control. (**E**–**H**) In a separate set of mouse tumor models, mice were administered i.p. with drug alone (50 mg/kg gemcitabine or 10 mg/kg GSK-126) or gemcitabine plus drug (10 mg/kg of GSK-126, SB-269970, or telotristat) for 38 days (*n* = 6). Tumor volume (**E**), gross appearance of tumor in mice (**F**), and tumor weight (**G**) at the time of sacrifice, and mouse body weight during the treatment period (**H**). * *p* < 0.05 compared to the vehicle-treated control. ^#^
*p* < 0.05 compared to GSK-126-treated group.

**Figure 8 cancers-13-05305-f008:**
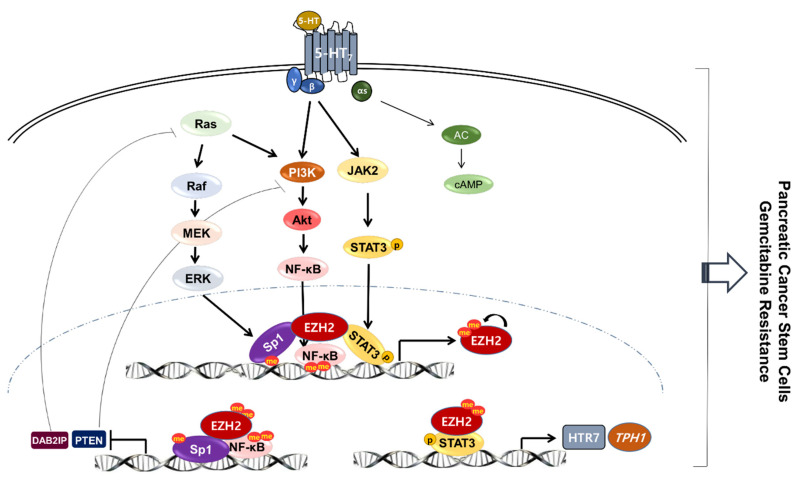
Schematic summary of noncanonical permissive action of EZH2 on TPH1-5-HT_7_ expression in PDAC. 5-HT activates PI3K/Akt and JAK2/STAT3 through Gβγ linked to 5-HT_7_. Together with Ras-activated ERK signaling, these signaling pathways enhance the nuclear level of Sp1, NF-κB, and p-STAT3, leading to EZH2 upregulation. Under the permissive action of EZH2, p-STAT3 is required for upregulation of TPH1 and 5-HT_7_, whereas Sp1 and NF-κB contribute to transcriptional repression of DAB2IP and PTEN. Through its noncanonical action, EZH2 activates intracellular signaling pathways through two ways, (1) activation of PI3K/Akt and JAK2/STAT3 through transactivation of TPH1-5-HT_7_ expressions, and (2) maintenance of the PI3K/Akt activity by inhibition of its inhibitory signals through downregulation of DAB2IP and PTEN.

## Data Availability

The data presented in this study are available in this article and Appendix A.

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
