# Peer review of "TPH1 and 5-HT7 Receptor Overexpression Leading to Gemcitabine-Resistance Requires Non-Canonical Permissive Action of EZH2 in Pancreatic Ductal Adenocarcinoma"

_cancers, 2021, doi:10.3390/cancers13215305_

Round 1

Reviewer 1 Report

The overall goal of study by Chaudhary et  was to determine if the serotonin (5-hydroxytryptamine, 5-HT) system plays  a role in gemcitabine resistance and the maintenance of pancreatic cancer stem cells (CSCs) in pancreatic cancer disease. Authors also tested if there an association of resistance emergence with Enhancer of zeste homolog 2 (EZH2), an epigenetic regulator of transcription. In this manuscript, authors conducted experiments to investigate the regulatory mechanisms underlying overexpression of EZH2, tryptophan hydroxylase 1 (TPH1), and 5-HT7, vis-a-vis gemcitabine resistance and pancreatic cancer stem cell survival. This is a well-conceived study. The experiments performed well and  the data interpretation is appropriate matching the results. The manuscript is written very well and in a reader friendly manner. The supplementary data provided shows that the study has been repeated and data is reproducible.  Following are some concerns:

  1. Tumors extracted from the mice  are not validated for the molecular pathway (TPH1, EZH2, HT7) by immunoshisopathology.
  2. Mice received intraperitoneal injection of drugs everyday for 42 days. It raises a question if animals faced any systemic toxicity as well as localized necrosis at the injection site. The whitish necrotic area and inflammation are common in mice if injected everyday, and that too  for  42 days. Authors need to provide evidence if the IACUC authorized for such a schedule of therapy.
  3. Authors should provide the LFT and KFT profile of mice to ensure if the therapy regimen and scheduling are safe.

Author Response

Reviewer 1

Overall goal of study by Chaudhary et al was to determine if the serotonin (5-hydroxytryptamine, 5-HT) system plays a role in gemcitabine resistance and the maintenance of pancreatic cancer stem cells (CSCs) in pancreatic cancer disease. Authors also tested if there an association of resistance emergence with Enhancer of zeste homolog 2 (EZH2), an epigenetic regulator of transcription. In this manuscript, authors conducted experiments to investigate the regulatory mechanisms underlying overexpression of EZH2, tryptophan hydroxylase 1 (TPH1), and 5-HT7, vis-a-vis gemcitabine resistance and pancreatic cancer stem cell survival. This is a well-conceived study. The experiments performed well and the data interpretation is appropriate matching the results. The manuscript is written very well and in a reader friendly manner. The supplementary data provided shows that the study has been repeated and data is reproducible.

--> Thank you for the review comments.

Following are some concerns:

  1. Tumors extracted from the mice are not validated for the molecular pathway (TPH1, EZH2, HT7) by immunohistopathology.

--> The tumors resected from mice are PANC-1 cell mass which were implanted subcutaneously and grown in mice.

In this study, we demonstrated that PDAC cells, including PANC-1 cells and MIA PaCa-2 cells, exhibited a positive correlation between gemcitabine resistance and overexpression of EZH2, TPH1, and 5-HT7. We also demonstrated the molecular pathway of EZH2-TPH1-5-HT7 axis by knockdown of genes in the axis and using specific chemical inhibitors (GSK-126, telotristat, and SB-269970) in PANC-1 cells and MIA PaCa-2 cells. Additionally, we validated the regulatory mechanism of EZH2-dependent TPH1 and 5-HT7 expression in PANC-1 and MIA PaCa-2 cells.

Verification of the potent anti-tumor activity of the inhibitors specific to EZH2-TPH1-5-HT7 axis is the very way to confirm and validate the molecular pathway of EZH2-TPH1-5-HT7 axis. Therefore, we think that immunohistopathology of TPH1, EZH2, 5-HT7 in the excised tumor tissues may not be necessary for the overall verification of the role of EZH2-TPH1-5-HT7 axis in PDAC cancer resistance and CSC maintenance.

  1. Mice received intraperitoneal injection of drugs every day for 42 days. It raises a question if animals faced any systemic toxicity as well as localized necrosis at the injection site. The whitish necrotic area and inflammation are common in mice if injected everyday, and that too for 42 days. Authors need to provide evidence if the IACUC authorized for such a schedule of therapy.

-->  We treated mice once a day, 6 days per week. In the two independent experiments of mouse tumor model, there were no signs considered to be systemic toxicity. Although mild scar reflecting inflammation was found intermittently at the injection site, but the scar disappeared after 2-3 days, and there was no case of necrosis throughout the experiments.

I have attached here IACUC approval certificates of the studies. (Please find the response letter of PDF file attached)

  1. Authors should provide the LFT and KFT profile of mice to ensure if the therapy regimen and scheduling are safe.

-->  We did not check LFT and KFT profile of the mice, because we had not noticed any systemic toxicity of the drug administration. Also, there was no significant body weight dropping in two series of experiments. As proof of no systemic toxicity, I have included here body weight and photo images of the mice at the time of sacrifice. (Please find the response letter of PDF file attached)

Reviewer 2 Report

The article by Chaudhary et al elucidated the regulatory role of EZH2-TPH1-5-HT-5-HT7 axis in imparting gemcitabine resistance and increasing CSC survival in PDAC. The authors have shown that inhibition of this axis led to decrease in CSC populations and recovery from gemcitabine resistance in PANC-1 and MIAPaCa-2 cells, whereas 5-HT treatment induced gemcitabine resistance in Capan-1 cells via increased expression of EZH2, TPH1, and 5-HT7. They further shown that PI3K/Akt and JAK2/STAT3 signaling pathways are implicated in the EZH2-TPH1-5-HT-5-HT7 axis driven gemcitabine resistance. Genetic or pharmacologic inhibition of EZH2 were shown to downregulate TPH1 and 5-HT7, inhibit PI3K/Akt and JAK2/STAT3 signaling pathways and alter nuclear levels of NF-kB, Sp1, and pSTAT3. Finally, they had shown the use of the inhibitors of EZH2, TPH1, or 5-HT7 in regressing tumor growth in a PANC-1 xenograft model. Overall, this is an excellent study with well-designed experiments and the conclusions are supported by the presented data. There are only a few minor concerns.

  1. EZH2 siRNA downregulated EZH1 significantly in addition to EZH2 downregulation (Fig. 1D). The authors should use a more specific siRNA for EZH2 knockdown.
  2. Did the authors check whether prolonged exposure of Capan-1 cells to 5-HT increased pancreatic CSC population?

Author Response

Reviewer 2

The article by Chaudhary et al elucidated the regulatory role of EZH2-TPH1-5-HT-5-HT7 axis in imparting gemcitabine resistance and increasing CSC survival in PDAC. The authors have shown that inhibition of this axis led to decrease in CSC populations and recovery from gemcitabine resistance in PANC-1 and MIAPaCa-2 cells, whereas 5-HT treatment induced gemcitabine resistance in Capan-1 cells via increased expression of EZH2, TPH1, and 5-HT7. They further shown that PI3K/Akt and JAK2/STAT3 signaling pathways are implicated in the EZH2-TPH1-5-HT-5-HT7 axis driven gemcitabine resistance. Genetic or pharmacologic inhibition of EZH2 were shown to downregulate TPH1 and 5-HT7, inhibit PI3K/Akt and JAK2/STAT3 signaling pathways and alter nuclear levels of NF-kB, Sp1, and pSTAT3. Finally, they had shown the use of the inhibitors of EZH2, TPH1, or 5-HT7 in regressing tumor growth in a PANC-1 xenograft model. Overall, this is an excellent study with well-designed experiments and the conclusions are supported by the presented data.

--> Thank you for the review comments.

There are only a few minor concerns.

  1. EZH2 siRNA downregulated EZH1 significantly in addition to EZH2 downregulation (Fig. 1D). The authors should use a more specific siRNA for EZH2 knockdown.

--> In this study, we have used MISSION® esiRNA (100 nM) from Merck, Sigma-Aldrich Solutions. The esiRNA are comprised of a heterogeneous pool of siRNA (natural RNA, no modifications) that all target the same mRNA sequence. The vendor guaranteed that these multiple silencing triggers lead to highly specific and effective gene knockdowns with lower off-target effects than single, chemically-synthesized siRNA. However, despite the guarantee, there was a decrease in EZH1 expression level by EZH2 esiRNA. This may be due to their 63% overall identity and 94% identity of their SET domain.

EZH1 is a paralog of EZH2 and likely arises from an EZH2 gene duplication. There is a switch between EZH1 and EZH2 in different biological contexts, quiescence versus proliferation, respectively. EZH1 is widely expressed in virtually all adult cells, whereas EZH2 is preferentially expressed in proliferating cells but barely expressed in terminally differentiated cells. Because the present study investigated the role of EZH2 in drug resistance and cancer stem cell maintenance in EZH2-overexpressing pancreatic cancer cells, we believe that the slightly lower specificity of EZH2 esiRNA may not be a significant obstacle in proving the overall subject.

  1. Did the authors check whether prolonged exposure of Capan-1 cells to 5-HT increased pancreatic CSC population?

-->  Yes, we did. Sphere forming ability of Capan-1 cells which were pretreated with 5-HT (10 μM) for 96 h was significantly increased. The result is now shown in Figure 1I. In the main text, “In addition, prolonged exposure of Capan-1 cells to 5-HT significantly enhanced sphere formation in Capan-1 cells (Figure 1I)”is added in page 5, lines 217-218.
